# Is There a Place for Cannabinoids in Asthma Treatment?

**DOI:** 10.3390/ijms26073328

**Published:** 2025-04-02

**Authors:** Agata Anna Lewandowska, Cezary Rybacki, Michał Graczyk, Dorota Waśniowska, Małgorzata Kołodziej

**Affiliations:** 1Clinical Department of Pulmonology, Allergology and Pulmonary Oncology, 10th Military Clinical Hospital with Polyclinic in Bydgoszcz, 85-681 Bydgoszcz, Polandcezary.rybacki@pbs.edu.pl (C.R.);; 2Faculty of Medicine, Bydgoszcz University of Science and Technology, 85-796 Bydgoszcz, Poland; 3Department of Palliative Care, Collegium Medicum in Bydgoszcz, Nicolaus Copernicus University in Toruń, 87-100 Toruń, Poland; 4Clinical Department of Oncology, Oncology Center of Professor Franciszek Łukaszczyk in Bydgoszcz, 85-796 Bydgoszcz, Poland

**Keywords:** cannabinoids, cannabis, CBD, THC, asthma, anti-inflammatory, bronchoconstriction, allergy

## Abstract

The beneficial effects of cannabinoids in the treatment of respiratory diseases have been drawing researchers’ attention for several decades. Asthma is a complex disease entity characterized by a variable course, the treatment of which requires the continuous search for alternative, adjuvant treatment strategies designed for patients refractory to available pharmacotherapies. Cannabinoids exert certain physiological responses in the respiratory system due to their immunomodulatory properties and the strong presence of the endocannabinoid system in the lungs. In animal model studies, THC and CBD seem to counteract bronchoconstriction and inhibit pro-inflammatory mediation, respectively, which highlights their possible future contribution to the treatment of respiratory and allergic diseases, such as asthma. However, there are controversies regarding the health consequences of cannabis usage, the extracts’ proportions, or equally safe and effective routes of administration, especially considering the alarming reports indicating an increased risk of asthma development among recreational cannabis smokers. The purpose of this review is to analyze the available literature on the influence of the endocannabinoid system, phytocannabinoids, and their modes of action on asthma pathogenesis in an attempt to assess their potential clinical relevance and determine future research directions.

## 1. Introduction

The term “cannabis” is used to define the natural products derived from the plant *Cannabis sativa* L., which contains a mixture of over 500 isolated compounds, such as cannabinoids, phenols, flavonoids, terpenoids, alcohols, wax esters, or alkaloids [1,2]. Cannabidiol (CBD) and Δ9-tetrahydrocannabinol (THC) constitute the most studied and recognizable phytocannabinoids known for their medicinal properties; however, there are many others [1,3]. CBD, considered a non-psychotropic compound, exerts anti-inflammatory, anti-oxidative, anxiolytic, and neuroprotective effects [4,5,6]. The therapeutic potential of THC, generally associated with psychoactive properties, mainly involves analgesia, muscle relaxation, and cessation of nausea [6].

The endocannabinoid system (ECS) is responsible for the mediation of physiological effects involved in the regulation and maintenance of cellular homeostasis [7,8]. It consists of the two key G protein-coupled cannabinoid receptors, CB1 and CB2, endogenously produced cannabinoids, anandamide (AEA) and 2-arachidonoylglycerol (2-AG), and the enzymes responsible for their synthesis and degradation, fatty acid amide hydrolase (FAAH) and monoacylglycerol lipase (MAGL) [9,10,11]. Phytocannabinoids act selectively on the ECS with the capacity to modulate the functions of both central and peripheral processes [8]. There are also synthetic cannabinoids, which constitute a group of artificially synthesized substances that exhibit agonist, antagonist, or inverse agonist activity toward the cannabinoid receptors [11].

CB1 is mainly expressed in the central nervous system; however, its presence is detected in several other organs, such as muscles, the liver, the pancreas, and adipose tissue [4,12]. CB2 is majorly expressed on immune cells, such as neutrophils, monocytes, NK cells, as well as lymphocytes B, CD4+, and CD8+, where it exhibits strong immunomodulatory effects [4,12]. The properties of cannabinoids not only rely on the activity of cannabinoid receptors but also others, such as transient receptor potential vanilloid channels (TRPVs), serotonin 1A (5-HT_1A_) receptors, a class of peroxisome proliferator-activated receptors (PPARs), G protein-coupled receptors (GPR55, GPR3, and GPR5), and adenosine A2A and glycine receptors [1,4,13,14]. The administration of cannabis-based products is also associated with the entourage effect—a phenomenon based on the synergic, enhanced activity of the plant’s compounds, such as cannabinoids, terpenes, and flavonoids, if applied together [15,16]. However, there are no clinical trials able to validate the entourage effect, exhibit its variability, or assess the products’ consistency [15,16].

The impact of cannabis administration on lung function and respiratory disorders has been drawing researchers’ attention for several decades [11,17]. Evidence indicates a strong presence of the ECS in the human lung and bronchial tissue, with most cell types expressing cannabinoid receptors and the ability to modulate immune responses [17]. The complexity of modes of action justifies the variety of and, sometimes, inconsistency in the cannabis properties reported in the literature [14]. Nevertheless, considering the vast distribution of both CB1 and CB2 receptors in human leukocytes and airway epithelial cells, cannabis consumers are undoubtedly expected to experience certain physiological responses from the respiratory system [7,17].

The potential of cannabinoids to inhibit bronchoconstriction and modulate inflammatory processes highlights their possible future contribution to the treatment and prevention of respiratory diseases. However, there are still numerous challenges and controversies that need to be addressed, such as the long-term consequences of regular usage, defining equally safe and effective routes of administration, as well as the extracts’ proportions, and their optimal doses in particular disorders [6,8,18,19,20].

Based on the literature available in the PubMed, Scopus, and Web of Science medical databases, a thematic analysis was conducted on the influence of the ECS, phytocannabinoids, and their modes of action in the course of asthma pathogenesis. Keywords, including cannabis, cannabinoids, THC, CBD, asthma, allergic, respiratory, bronchodilator, anti-inflammatory, and vaporization, were used to source the data. Specifically, studies on animal models and in vitro using synthetic agents acting selectively on the cannabinoid receptors and other elements of the ECS, as well as phytocannabinoids, were chosen and carefully reviewed.

## 2. Asthma

Asthma is a disease characterized by chronic airway inflammation and hyperreactivity [14,21,22]. It affects more than 300 million people among the world’s total population, with the prevalence ranging from 1 to 29%, depending on the region [21,23,24,25]. Different phenotypes of asthma have been distinguished based on demographic, clinical, and pathophysiological characteristics, including allergic asthma, non-allergic asthma, late-onset asthma, asthma associated with obesity, and asthma with persistent airflow limitation [21].

In allergic (type 2) asthma, the pathogenesis is associated with T helper 2 (Th2) cells, a distinct lineage of CD4+ effector T-cells, as well as type 2 innate lymphoid cells (ILC2s), which have the capacity to release type 2 cytokines, such as IL-4, IL-5, IL-9, IL-13, and IL-31, resulting in B cell differentiation and a robust influx of eosinophils and immunoglobulin E (IgE) in the airways [22,23,26,27,28]. ILC2 cells can also be activated by IL-25, IL-33, and thymic stromal lymphopoietin (TSLP) [27]. The involvement of ILs associated with Th2-cell-mediated response is crucial in the pathogenesis of key asthma features, such as eosinophilia, elevated IgE serum level, airway hyperreactivity (AHR), and airway remodeling [26]. Allergen-specific IgE is considered the hallmark of atopy and allergic diseases, with a strong correlation between its production and the development and severity of type 2 asthma [29,30,31].

Non-allergic (non-type 2) asthma includes the neutrophilic and paucigranulocytic phenotypes [23,32]. The inflammatory processes are mediated by non-Th2 cytokines, such as IL-17, mostly produced by CD4+ Th17 cells, as well as tumor necrosis factor-α (TNF-α) and interferon-γ—both activated by Th1 cells [23,32,33].

AHR, a cardinal feature of asthma, is defined as the predisposition of the airways to narrow excessively in response to nonspecific stimuli in predisposed individuals [34]. The production of inflammatory cytokines by the airway epithelium can be triggered by pollution, injury, or infection, eventually leading to bronchoconstriction [21,22,23]. In asthma, the inflammatory processes additionally lead to mucus hypersecretion, increased vascular permeability, and edema [35]. There is still little consensus on the mechanisms underlying AHR; however, it is believed that indirect stimuli irritate airway smooth muscle cells via the activation of pro-inflammatory agents and the release of mast cell mediators, such as histamine, leukotriene D_4_, and prostaglandin D_2_ [23,27,35,36,37]. Direct stimulants induce twofold contraction—through the activation of muscarinic M3 receptors and the release of intracellular calcium ions (methacholine) or the activation of H1 receptors, resulting in bronchoconstriction via the vagal reflex (histamine) [38,39,40]. It is worth mentioning that the intensity of AHR should not be used as an index of inflammation in the airways, just as the sole presence of inflammatory cytokines in the respiratory tract is not sufficient to trigger bronchoconstriction [38].

The diagnosis of asthma is based both on the characteristic clinical presentation, including common symptoms such as cough, wheezing, and dyspnea, as well as the evidence of variable expiratory airflow limitation in bronchodilator reversibility testing, a positive bronchial/exercise challenge test, relevant variation in peak expiratory flow (PEF)/forced expiratory volume in 1 s (FEV1) monitoring, or a significant increase in FEV1 after the administration of inhaled glucocorticosteroids (GCSs) [21,22,23,41].

Inhaled GCSs remain the mainstay of treatment, additionally supported by long-acting beta-2 agonists, muscarinic receptor antagonists, leukotriene receptor antagonists, and rescue medicines in the form of short-acting beta-2 agonists, and oral GCSs—aimed at controlling severe asthma symptoms [41,42,43,44]. Asthmatic patients require regular, control-based management in order to sustain satisfactory quality of life, especially in the case of adult-onset asthma with neutrophilic inflammation, often associated with poor response to GCS therapy [25,42,45].

Severe asthma, associated with significant comorbidity and frequent healthcare resource use, is estimated to represent 5–10% of the entire asthma population [46]. Severe type 2 asthma is currently treated using several monoclonal antibodies approved in clinical practice, such as anti-IgE (omalizumab), anti-thymic stromal lymphopoietin (tezepelumab), anti-IL-5 (benralizumab and mepolizumab), and both anti-IL-4 and anti-IL-13 (dupilumab) [47,48,49,50]. However, no effective biologic therapy is available to date in non-type 2, neutrophilic asthma as its pathogenesis and inflammatory mediation are still poorly understood, which necessitates the search for alternative treatment options [47,51].

## 3. Cannabinoid Receptors

The bronchodilator activity of cannabinoids has been evaluated over the past few decades due to their fairly independent mode of action compared to the available bronchodilators used in obstructive respiratory disorders [52,53]. This mediation seems to depend majorly on the activation of CB1 receptors, distributed on the nerve fibers of bronchial and bronchiolar smooth muscle cells, which leads to the presynaptic inhibition of cholinergic-induced airway contraction [54]. Considering the strong affinity to the CB1 receptor, THC is believed to exert potential dilating effects on human airways; however, the significance of this phenomenon is still unclear, especially since some asthmatic patients react to its administration with a paradoxical bronchospasm [55,56,57]. The mechanism could be explained by the dual action of endogenous AEA on CB1 receptors, which seems to inhibit smooth muscle constriction triggered by an irritant but then again induces the bronchospasm after the constricting tone exerted by the vagus nerve is removed [55]. AEA is produced in the lung tissue upon calcium-ion stimulation, which suggests its involvement in the intrinsic control of AHR [55]. In addition to the above, the reported study confirmed the capacity of the CB1 antagonist (SR141716A) to enhance irritant-induced bronchospasm and cough, indicating the potential contribution of the selective cannabinoid-based agents in the treatment of obstructive respiratory diseases [55].

Another study investigated the influence of two synthetic cannabinoids, ACEA (CB1 agonist) and JWH133 (CB2 agonist), on 5-hydroxytryptamine-induced airway contraction in a non-atopic asthma model in mice [58]. The CB1 agonist, as opposed to the CB2 agonist, inhibited tracheal hyperreactivity; however, neither of them attenuated the increase in the macrophage number in bronchoalveolar lavage fluid (BALF) [58].

In an allergic asthma model, guinea pigs were challenged with ovalbumin and treated with both FAAH and MAGL inhibitors to assess airway inflammation and hyperreactivity [59]. The selective inhibition of FAAH (URB597) downregulated the production of pro-inflammatory cytokines and inflammatory cell infiltration but did not exert any effect on AHR [59]. However, the selective inhibition of MAGL (JZL184) and dual inhibition of FAAH and MAGL (JZL195) alleviated both the inflammatory response and AHR, suggesting a promising therapeutic strategy for allergic asthma [59]. Other reports have pointed to the FAAH-dependent AEA metabolites as important mediators of airway muscle relaxation both in vitro and in vivo [60].

The ECS is believed to influence the pathophysiology of asthma by modulating the activity of immune system cells [46]. It has been demonstrated that cannabinoids exhibit anti-inflammatory properties in the airways of human patients, while other reports suggest that they might severely exacerbate asthma and allergic diseases through CB2 receptor mediation [9]. Eosinophils and monocytes express large amounts of CB2 receptors, making them strongly responsive to cannabinoids [17]. The authors of one of the reports attempted to confirm the direct influence of CB2 receptors on human and mouse eosinophil effector function in vitro and in vivo [61]. The results indicated that the selective CB2 receptor agonist (JWH133) enhanced chemoattractant-induced eosinophil expression, shape change, chemotaxis, and the production of reactive oxygen species [61]. Such effects were absent in eosinophils derived from CB2 knockout mice or in the case of using the selective CB2 antagonist, therefore confirming receptor specificity [61]. Additionally, systemic administration of JWH133 aggravated both AHR and eosinophil influx into the airways [61]. There was no such effect in eosinophil-deficient mice, which again indicated that CB2 receptors directly contribute to the pathogenesis of eosinophil-driven diseases [61]. CB2 receptor antagonism has therefore been evaluated as a promising pharmacological strategy for the treatment of allergic diseases, such as asthma [61].

Similarly, the authors of another study investigated the signaling of oleoylethanolamide—an endogenous cannabinoid-like compound—which has been reported to be elevated in severe asthma and aspirin-exacerbated respiratory disease [62]. In mice, the expression of CB2 receptors in peripheral blood eosinophils and dEol-1 cells was increased after oleoylethanolamide administration [62]. Using a CB2 antagonist reduced the activity of peripheral blood eosinophils and dEol-1 cells and the level of pro-inflammatory and type 2 cytokines [62]. Additionally, AHR and eosinophil recruitment were alleviated [62].

ILC2s play a key role in type 2 asthma [63]. There is evidence that enhanced CB2 signaling in pulmonary ILC2s stimulated the development of ILC2-dependent AHR and lung inflammation both in mice and humans [63]. Conversely, the deprivation of CB2 signaling in ILC2s attenuated lung inflammation [63]. However, the mechanisms related to the influence of cannabinoids on the activation of ILC2s remain to be evaluated [63].

On the other hand, the authors of another study also examined the effect of the selective CB2 agonist (P6023) on allergic airway inflammation in a house dust mite-induced animal model of asthma [64]. Intranasal administration of P6023 attenuated the inflammatory process by inhibiting the accumulation of CD4+ T cells, eosinophils, and Th2 cytokines in the lungs [64]. Moreover, AHR and mucus hypersecretion were ameliorated, demonstrating the beneficial effects of CB2 agonists [64].

In non-type 2 inflammation, T helper cells 17 (Th17) prompt airway epithelial cells and fibroblasts to release neutrophil chemokines, such as IL-8, growth-related oncogene-alpha, and granulocyte macrophage stimulating factor, which aggravate the inflammatory infiltration of the airways and lungs [65]. On the other hand, regulatory T cells (Tregs) suppress the inflammatory response by modulating the Th1/Th2 balance and Th17 recruitment [65]. Studies based on animals have confirmed that Tregs also secrete immunosuppressive cytokines, such as IL-10 and transforming growth factor β, which lead to a reduction in inflammation, AHR, and remodeling [65,66,67]. A decreased number of Tregs compared to Th17 provokes the immunological pathogenesis of asthma, which is reflected in the reduced levels of Tregs in the lungs of asthma patients [65,68,69,70]. In an animal model of neutrophilic asthma, a CB2-selective agonist (β-caryophyllene) increased the proportion of Tregs to Th17 by promoting CD4+ T cell differentiation into Tregs but not Th17 [65]. As a result, the activation of CB2 receptors modulated the Tregs/Th17 balance, inhibited the activity of Th17, and reduced airway inflammation [65]. Despite the unclear mechanism of action, CB2 receptors seem to play a potential role in regulating the activity of Tregs, thus relieving the severity of non-type 2, neutrophilic asthma [65].

Another study evaluated the effects of the synthetic CB1/CB2 receptor agonist (CP55, 940) on antigen-induced asthma-like reactions in sensitized guinea pigs [71]. The ovalbumin challenge triggered abnormalities in respiratory, morphological, and biochemical parameters, such as the levels of prostaglandin D_2_ and TNF-α, which were later markedly reduced by CP55, 940, indicating the protective role of both CB1 and CB2 receptors on lung function [71]. Interestingly, pre-treatment with the selective CB1 and CB2 antagonists reverted these effects, thus highlighting the importance of the interactions between the cannabinoid receptors [71].

There is still much to be discovered in the understanding of the action of cannabinoids on the channels other than the cannabinoid receptors, such as the TRPV group, which constitutes a potential drug target for asthma due to its involvement in AHR, inflammation, and remodeling [72,73,74]. Interestingly, while TRPV2 may represent a novel positive biomarker of airway smooth muscle function and immune cell physiology, TRPV1 and TRPV4 seem to contribute to the development and exacerbation of asthma [73]. Additionally, PPARs, particularly PPAR-*γ*, play an important role in the modulation of type 2 inflammation, making them a possible target in the treatment of asthma [75]. It remains unclear why some physiological processes are mediated by PPARs for certain cannabinoids and not others, despite a similar ability to activate them [76]. The properties of PPARs and their affinity to the ECS are undoubtedly ambiguous and require further research [75,76]. Finally, the activation of 5-HT_2_ receptors is proposed as a novel treatment strategy for allergic asthma patients due to their potential to downregulate Th2 signaling, AHR, mucus production, and remodeling [77,78]. However, the contribution of cannabinoids to these processes has not yet been determined.

Collectively, the complex function of the ECS and selectively acting cannabinoids remain equivocal. It seems that in certain circumstances, the activation of CB2 receptors can demonstrate key importance in eosinophil recruitment and inflammation aggravation in the respiratory tract, suggesting that particular cannabinoid receptor-specific agents can be identified as pro-allergenic and able to induce the IgE-mediated response [9,14,61,79]. On the other hand, according to other reports, the cannabinoid receptors exhibit anti-inflammatory properties in the airways of allergic patients [9]. The effects exerted by cannabinoid receptor mediation in animal asthma models are presented in Table 1.

## 4. Phytocannabinoids

Older reports from the XX century have shown positive, rapid bronchodilator effects of inhaled and oral THC on asthmatic patients; however, the evidence is scarce and generally inconclusive, especially in the case of chronic exposure [57,80,81,82,83,84]. In guinea pigs, the intratracheal administration of TNF-α potentiated chemokine-dependent airway neutrophilia and vagal-induced contractions of the airway smooth muscle by enhancing postganglionic acetylcholine release [54]. According to the results of the study, only THC (but not CBD, cannabigerol, cannabichromene, or cannabidiolic acid) attenuated TNF-α-induced neuronal cholinergic transmission, therefore reducing bronchoconstriction through the presynaptic activation of both CB1 and CB2 receptors, as well as presenting anti-inflammatory and antitussive effects [54]. The outcome did not differ in the case of simultaneous use of CBD and THC [54].

Another study aimed to evaluate the potential of CBD in reversing airway inflammation and remodeling in an allergic model of Balb/c mice exposed to ovalbumin [14]. CBD administration decreased AHR, remodeling, the level of collagen fiber in both airway and alveolar septa, and the expression of pro-inflammatory markers in BALF [14]. The mechanisms appeared to be mediated by the interaction between both CB1 and CB2 receptors [14].

In rats sensitized to ovalbumin, CBD treatment downregulated the plasma levels of Th1 and Th2, as well as their allergen-associated cytokines, such as IL-4—involved in Th2 cell differentiation, IgE production, and eosinophil trafficking—and IL-5—crucial for the activation and recruitment of eosinophils [35,85,86]. Additionally, CBD demonstrated the potential to decrease the levels of TNF-α, the major mediator of severe asthma, IL-6, a cytokine stimulating T cell proliferation, and IL-13, involved in every aspect of asthma pathophysiology, such as IgE production, eosinophil recruitment, maturation of mucus-secreting goblet cells, and enhanced airway smooth muscle contractility [85,87].

Another study examined the influence of high-concentration CBD extract on the activity of human immune cells engaged in asthma pathogenesis [45]. CBD inhibited the differentiation of CD4+ T cells into Th2 cells, crucial in type 2 asthma aggravation due to their secretion of IL-5 and IL-13, and therefore mitigated the Th2-mediated immune response and eosinophilic inflammation [45]. Additionally, in human-derived neutrophils, CBD downregulated the release of non-type 2 asthma cytokines, such as IL-8 and IL-6, which resulted in impaired neutrophil migration [45]. These conclusions were reflected in the animal asthma model (ovalbumin-treated mice), as CBD therapy led to a decrease of IgE, IL-4, IL-5, and IL-13 levels in both blood and lung tissue, as well as eosinophil and neutrophil infiltration in the lung [45]. Therefore, high-concentration CBD extracts seem to be a potentially promising adjunctive treatment for both type 2 and non-type 2 asthma [45].

Collectively, in animal asthma models, CBD and THC exhibit consistent anti-inflammatory, bronchodilator, and antitussive properties, as shown in Table 2. Despite the relatively numerous studies presenting promising results in animal models, the clinical significance of phytocannabinoids’ activity on the respiratory system and asthma in humans remains insufficient and poorly understood [46,54,88]. A better understanding of the mechanisms of action exerted by specific compounds and extract formulations is warranted in the context of designing potential novel treatment strategies [9,89].

## 5. Terpenes

Terpenes constitute a diverse class of naturally occurring components of cannabis resins and essential oils, which are associated with anti-inflammatory, anti-oxidative, and analgesic properties, used in modern medicine [90,91]. Together with cannabinoids, they are believed to exert synergic and/or additive effects [92]. Several terpene compounds, such as α-phellandrene, 1,8-cineole, α-pinene, β-pinene, borneol, linalool, humulene, and terpinen-4-ol, have been evaluated for their therapeutic potential in the treatment of asthma and other lung injury models [93,94,95,96,97,98,99]. For instance, 1.8-cineole, a monoterpene known for its mucolytic and spasmolytic action on the respiratory tract, inhibited the production of leukotriene B4 and prostaglandin E2 in blood monocytes of asthmatic patients [93,100]. Moreover, 1,8-cineole downregulated the levels of pro-inflammatory cytokines, such as IL-1β, IL-4, IL-6, IL-13, IL-17A, and TNF-α, in the BALF of animal asthma models sensitized with ovalbumin, house dust mite, and cigarette smoke [95,101,102,103]. Borneol and terpineol inhibited histamine-induced bronchoconstriction of tracheal smooth muscles in guinea pigs in vitro [95]. Finally, α-pinene has been demonstrated as a bronchodilator in human volunteers, also suggesting its potential clinical relevance in the treatment of asthma [95,104].

## 6. Recreational Use

The legalization of recreational cannabis smoking in many countries has led to an increase in its usage, which seems to be associated with harmful and underestimated health consequences for both healthy individuals and patients suffering from chronic lung diseases [105,106,107,108,109,110]. It is estimated that 1 out of 20 people aged 15–64 years old uses cannabis recreationally, which represents 2.6–5% of the world’s adult population [105].

Health consequences associated with cannabis usage arise primarily from the harmful chemicals produced during the combustion of the cannabis plant, as well as the reduced bioavailability of the original ingredients [1,111,112]. Similarly to the effects of cigarettes, smoking cannabis aggravates inflammation in the lung, expressed by the elevated number of neutrophils and macrophages, with their functional impairment [111,113,114]. Additionally, cannabis smoke, like tobacco smoke, is believed to worsen respiratory symptoms, such as cough, wheezing, and dyspnea [111,115,116,117,118]. It increases the risk of airway obstruction and exacerbation of lung diseases; however, the evaluation is difficult to untangle due to the common concurrent use of tobacco [65,116,119,120,121]. Interestingly, there is no documented association between cannabis usage and the development of chronic obstructive pulmonary disease (COPD) [105,121,122].

Patients with asthma and allergies are considered a high-risk group for the adverse effects of cannabis [108,109,123,124]. A systematic review and meta-analysis examined the incidence of asthma among cannabis users and provided disturbing evidence that they present a 31% higher risk of having asthma compared to non-users [117]. Data drawn and analyzed from the 2020 National Survey on Drug Use and Health also indicated a positive linear relationship between the frequency of cannabis use and current asthma prevalence among US individuals, adjusting for demographics and cigarette smoking [125]. Another retrospective study enrolled 406,800 patients who presented to hospital with acute asthma exacerbation, 16,915 of whom admitted to concurrent cannabis usage [126]. Compared to patients with no cannabis use, there was a higher risk of mechanical ventilation, altered mental status, and mortality rate [126].

In the case of the respiratory system, oral and inhalatory administration of cannabis are acceptable [1,112,127,128,129]. As opposed to smoking, the technique of vaporization is based on heating the extracts to a temperature below the point of their combustion—approximately 160–230 degrees Celsius—which creates vapor with reduced levels of toxins, such as benzene, toluene, tar, and ammonia [112,130,131]. Vaporization reduces the risk of respiratory complications occurring in the case of cannabis combustion; however, the available vaping devices vary in safety [132]. Based on toxin exposure, metered dose inhalers and dried product vaporizers appear to demonstrate the lowest level of risk [132]. Despite the possibility of vaporizing cannabis, some patients still prefer smoking, often due to the concurrent tobacco habit or economic reasons [133,134]. The perceived cost advantage is, in fact, illusory, as the product runs out faster in the case of combustion, making smoking the costlier form of consumption [133,134].

Research based on the 2017–2019 Behavioral Risk Factor Surveillance System in the United States evaluated the prevalence and trends of cannabis vaping with a sample of 160,000 and demonstrated that individuals who vaped cannabis were more likely to concurrently vape nicotine, consume alcohol, and exhibit other high-risk behaviors [135]. However, no correlation with asthma or other respiratory symptoms was observed [135]. Additionally, unregulated vaping products invading the market worldwide are associated with the emergence of e-cigarette- and vaping-associated lung illness (EVALI); however, the health risks appear to result from contaminated and illicitly sourced vaping devices rather than cannabis itself [132,136]. In areas under the federal control of THC, no positive association between EVALI and cannabis vaporization products was observed [132,136]. Nevertheless, patients in whom inhalation proves to cause negative respiratory symptoms could benefit from oral extract preparations in capsule form or transmucosal sublingual products, depending on their regional availability [137,138]. Cannabis oil preparations not only facilitate easy dose modulation and high bioavailability of the active components due to the formulation’s lipophilicity but also offer a safe replacement for the inhalation route [139].

The general lack of knowledge about vaporization techniques, terminology, and different devices among both consumers and healthcare professionals carries implications for the safe and efficient use of cannabis products [132,140]. It is worth remembering that cannabis does not consist of a single compound, and there are many different formulations containing THC, CBD, terpenes, and others [129]. The preparations have altered significantly over the past decades, with the majority of older studies conducted with the use of THC only (devoid of the anti-inflammatory influence of CBD), making the conclusions discordant with the currently available strains [129]. Additionally, given that most human studies are based on recreational cannabis users who smoke the product, the existing evidence does not allow firm conclusions to be drawn about the effects of cannabinoids on the respiratory system, especially in the case of recommended, controlled routes of administration [20,56,128,141]. Nevertheless, smoking cannabis seems to be a risk factor for asthma development, and its usage should undoubtedly rely on a different route of application, if indicated [106,142]. Well-designed clinical trials on the pulmonary consequences of cannabis usage, vaping device types, and their safe application in respiratory disorders are still necessary in order to avoid misguided conclusions, especially given the complexity of overlapping mechanisms involved in asthma pathogenesis and cannabinoid activity [121,132,143]. Finally, education of medical professionals, and ultimately patients themselves, could also help to reduce the stigma associated with the recreational use of cannabis [144].

## 7. Conclusions

The ECS appears to significantly affect the respiratory system, especially in the case of inflammatory and allergic diseases. Asthma is a globally occurring, multifaceted disease entity characterized by a variable course and complex inflammatory mediation, which strongly encourages further research to identify potential adjuvant treatment strategies for patients refractory to the available therapeutic agents. Despite the sometimes conflicting results regarding the effects of selectively activated cannabinoid receptors, CB2 in particular, reports demonstrating the activity of phytocannabinoids have proven much more consistent. Both CBD and THC seem to exert beneficial therapeutic effects in animal asthma models in the form of anti-inflammatory and bronchodilator responses, respectively. These observations once again highlight the diversity of the mechanisms of action of cannabinoids and the importance of interactions between them. In addition to the studies demonstrating the mediation via the cannabinoid receptors, research focused on the alternative modes of action of cannabinoids in asthma is required. Phytocannabinoids, both CBD and THC, could potentially find application in the treatment of type 2 and non-type 2 asthma; however, they should be approached with caution, given the disturbing data indicating an increased risk of asthma development among recreational cannabis smokers. Additionally, the potential clinical relevance of THC in counteracting bronchoconstriction, however promising, remains uncertain and controversial, especially in the case of chronic exposure. Perhaps in the future, the adjuvant use of phytocannabinoids in the treatment of asthmatic patients will be justified after a carefully conducted, individual risk–benefit assessment. Currently, further high-quality research on the safest routes of administration and long-term health consequences is necessary.

## Figures and Tables

**Table 1 ijms-26-03328-t001:** The effects of the cannabinoid receptor mediation on the respiratory system in animal asthma models.

Mode of Action	Agent	Result	Clinical Implication	Reference
CB1 antagonism	SR141716	↑ Irritant-induced smooth muscle contraction	Exacerbation of bronchoconstriction and cough	[55]
CB1 agonism	ACEA	↓ 5-hydroxytryptamine-induced smooth muscle contraction	Prevention of tracheal hyperreactivity	[58]
FAAH/MAGL inhibition	JZL195	↓ Pro-inflammatory cytokine production ↓ Inflammatory cell infiltration	Alleviation of airway hyperreactivity and inflammation in allergic asthma	[59]
CB2 agonism	JWH133	↑ Chemoattractant-induced eosinophil shape change, chemotaxis, and CD11b surface expression ↑ Eosinophil influx in the airways ↑ Production of reactive oxygen species	Aggravation of airway hyperreactivity	[61]
CB2 antagonism	SR144528	↓ Activity of peripheral blood eosinophils and dEol-1 cells ↓ Level of inflammatory and type 2 cytokines	Alleviation of airway hyperreactivity	[62]
CB2 agonism	JWH133	↑ ILC2 proliferation and function ↑ ILC2-driven lung inflammation	Exacerbation of airway hyperreactivity	[63]
CB2 agonism	P6023	↓ Accumulation of CD4+ T cells and eosinophils ↓ Th2 cytokine production	Mitigation of airway hyperreactivity and mucus production	[64]
CB2 agonism	β-caryophyllene	↓ Infiltration of inflammatory agents (eosinophils, neutrophils, lymphocytes, IL-6, IL-8, and TNF-α) ↑ CD4+ differentiation into Treg cells ↑ Cytokines secreted by Tregs (TGF-β and IL-10) ↓ Cytokines secreted by Th17 (IL-17A and IL-22)	Amelioration of neutrophilic asthma symptoms	[65]
CB1/CB2 agonism	CP55, 940	↓ Leukocyte and eosinophilic infiltration ↓ Mast cell activation ↓ Free radical-induced DNA injury ↓ Myeloperoxidase activity ↓ TNF-α and prostaglandin D_2_ levels in BALF ↓ Bronchial lumen restriction and alveolar hyperinflation	Reduction in cough and dyspnea	[71]

↑, increased; ↓, decreased; CB1, cannabinoid receptor 1; FAAH, fatty acid amide hydrolase; MAGL, monoacylglycerol lipase; CB2, cannabinoid receptor 2; ILC2, type 2 innate lymphoid cells; IL, interleukin; TNF-α, tumor necrosis factor-α; Tregs, regulatory T cells; TGF-β, transforming growth factor β; BALF, bronchoalveolar lavage fluid.

**Table 2 ijms-26-03328-t002:** Anti-inflammatory, bronchodilator, and antitussive effects of phytocannabinoids in animal asthma models.

Phytocannabinoid	Result	Clinical Implication	Reference
THC	↓ TNF-α-enhanced vagal-induced bronchoconstriction ↓ Citric acid-induced cough response	Decrease in airway hyperreactivity and cough	[54]
CBD	↓ Collagen fiber content in airway and alveolar septa ↓ Proinflammatory markers in BALF and lung homogenate	Decrease in airway hyperreactivity and remodeling processes	[14]
CBD	↓ Th1 cytokines (TNF-*α* and IL-6) ↓ Th2 cytokines (IL-4, IL-5 and IL-13)	Potential adjunctive asthma treatment	[85]
High-concentration CBD extract	↓ Leukocyte, neutrophil, and eosinophil migration ↓ Differentiation of CD4+ T cells into Th2 cells ↓ Th2-mediated immune response ↓ IgE blood levels ↓ Secretion of type 2 cytokines (IL-4, IL-5, and IL-13) ↓ Secretion of non-type 2 cytokines (IL-8 and IL-6)	Potential adjunctive treatment for both type 2 and non-type 2 asthma	[45]

↓, decreased; THC, Δ9-tetrahydrocannabinol; TNF-α, tumor necrosis factor-α; CBD, cannabidiol; BALF, bronchoalveolar lavage fluid; IL, interleukin.

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
