# Peer review of "Is There a Place for Cannabinoids in Asthma Treatment?"

_ijms, 2025, doi:10.3390/ijms26073328_

Round 1
Reviewer 1 Report
Comments and Suggestions for Authors
The manuscript was well written and provide relevant discussion and informations about the cannabinoids use on asthma physiopatology.
In the current state, the article can be accepted posterior minor corrections:
- Clarify the article purpose presented on abstract, since its similar to a experimental article instead a review.
- Review the setence of lines 32 - 34. The suitable term is "cannabinoids" and not "cannabis".
- Provide a complete review regarding the abbreviation use.
- Insert details concerning the search methodology and criteria to select the studies.
- The critical point of view of the authors related to the recreational use and its disadvantages can be strengthen added informations about the pharmaceutical dosage forms, marjority oily solution, of CBD that provide a safe and effective use.
Author Response
Thank you for the favorable and thorough review of the manuscript.
I hope we have addressed all of the comments appropriately:
- The abstract was modified according to your suggestion to fit the article type better (changes marked in red).
- The sentence was changed in order to properly define the term “cannabis” – lines 32-34.
- The review regarding the abbreviation use was completed.
- Details concerning the search methodology and criteria to select the studies were included (lines 79-86).
- Information about the available dosage forms, which can offer both safe and effective administration, was added (lines 379-384).
Reviewer 2 Report
Comments and Suggestions for Authors
This manuscript reports a review of the knowledge relied on the cannabinoids and their receptors on the asthma treatment, but also on their contribution to asthma generation.
The manuscript is well written and the most relevant aspects are critically reviewed.
I have no much to say. But only I can suggest to add some comments where appropriate on the meaning of ILs and IgE regarding asthma causes and consequences.
Author Response
Thank you for such a favorable review of the manuscript.
According to your suggestion, we have added the comments regarding the importance of ILs and IgE on asthma pathogenesis and severity (lines 99-104).